# A Review of Thin-Film Magnetoelastic Materials for Magnetoelectric Applications

**DOI:** 10.3390/s20051532

**Published:** 2020-03-10

**Authors:** Xianfeng Liang, Cunzheng Dong, Huaihao Chen, Jiawei Wang, Yuyi Wei, Mohsen Zaeimbashi, Yifan He, Alexei Matyushov, Changxing Sun, Nianxiang Sun

**Affiliations:** 1Department of Electrical and Computer Engineering, Northeastern University, Boston, MA 02115, USA; liang.xi@husky.neu.edu (X.L.); dong.cu@husky.neu.edu (C.D.); chen.huai@husky.neu.edu (H.C.); wangjiawei07@gmail.com (J.W.); wei.yuy@husky.neu.edu (Y.W.); zaeimbashi.m@husky.neu.edu (M.Z.); he.yi@husky.neu.edu (Y.H.); matyushov.a@husky.neu.edu (A.M.); sunchangcing@163.com (C.S.); 2College of Science, Zhejiang University of Technology, Hangzhou 310023, China

**Keywords:** magnetoelastic materials, magnetostriction, magnetoelectric devices, thin films

## Abstract

Since the revival of multiferroic laminates with giant magnetoelectric (ME) coefficients, a variety of multifunctional ME devices, such as sensor, inductor, filter, antenna etc. have been developed. Magnetoelastic materials, which couple the magnetization and strain together, have recently attracted ever-increasing attention due to their key roles in ME applications. This review starts with a brief introduction to the early research efforts in the field of multiferroic materials and moves to the recent work on magnetoelectric coupling and their applications based on both bulk and thin-film materials. This is followed by sections summarizing historical works and solving the challenges specific to the fabrication and characterization of magnetoelastic materials with large magnetostriction constants. After presenting the magnetostrictive thin films and their static and dynamic properties, we review micro-electromechanical systems (MEMS) and bulk devices utilizing ME effect. Finally, some open questions and future application directions where the community could head for magnetoelastic materials will be discussed.

## 1. Introduction

### 1.1. Multiferroic Materials

Multiferroic materials are the materials that inherently exhibit two or more ferroic properties, such as ferroelectricity, ferromagnetism and ferroelasticity, etc. More recently, both single phase multiferroic materials and multiferroic composites have attracted intense interests due to the realization of strong magnetoelectric (ME) coupling, i.e., the control of electric polarization (P) by applying magnetic field (H) (direct ME effect), or the manipulation of magnetization (M) through electric field (E) (converse ME effect) [1,2,3,4,5,6,7,8]. Exciting progress has been made on novel multiferroic materials and multifunctional devices because of their high-performance ME coupling [9,10,11,12,13,14]. Based on the operating mechanisms for controlling different orders, multiferroic devices can be classified as the following groups listed in Table 1. Applications, such as sensors, energy harvesters, etc. [15,16,17,18,19,20,21,22,23,24,25] have been developed according to the direct ME coupling, while converse ME coupling has been utilized to design voltage tunable devices including ME random access memory (MERAM), inductors, etc. [26,27,28,29,30,31,32,33,34,35,36]. Besides, another promising topic that has received the significant research interests during recent years is mechanically actuated ME antennas, which exploit both direct and converse ME couplings [37,38,39,40,41,42,43,44,45,46]. The combined high permeability μ and permittivity ε offer great beneficial in compact RF/microwave devices, for example, miniaturized antennas, etc. [47,48]. Compared to conventional magnetic devices, these electric field tunable multiferroic devices have the advantages of lightweight, low power-consuming, compact, etc. For example, an integrated magnetic inductor based on solenoid structure with FeGaB/Al_2_O_3_ multilayer films was reported by Gao et al. [35]. A high quality factor and >100% inductance enhancement compared with that of the same size air core inductor across a wide frequency band of DC-2.5 GHz were achieved. Nan et al. [44] proposed the acoustically actuated ME antennas with a released ferromagnetic/piezoelectric thin-film heterostructures, which could miniaturize the antenna size in 1-2 orders without performance degradation over conventional compact antennas.

In order to achieve large tunability in multiferroic devices, strong ME coupling is required in the heterostructures. The strength of ME coupling can be described by two coefficients: $\alpha_{Direct}=\partial P/\partial H$ (direct ME effect) and $\alpha_{Converse}=\partial M/\partial E$ (converse ME effect). Therefore, multiferroic materials are of great importance in determining the performance of these multiferroic devices listed in Table 1. The ME effect was proposed by Curie in 1894 [49] and was proved by Landau and Lifshitz on the basis of the crystal symmetry more than 60 years later [50]. Dzyaloshinskii experimentally demonstrated the ME effect in single-phase multiferroic material Cr_2_O_3_ and intense research efforts have been made to explore the possibility of achieving ferroelectric and magnetic orders in a single-phase material since then. However, the single-phase materials, such as BiFeO_3_, BiMnO_3_, etc. are still suffering from the low Curie temperature and weak ME coupling coefficients [51]. By means of utilizing strain-mediated ME composites, which are composted of ferroelectric and magnetoelectric phases, a giant ME coupling even above room temperature can be obtained. The first work on combining piezoelectric and magnetostrictive effects was presented by Van Suchtelen in 1972 [52]. The reported ME voltage coefficient of BaTiO_3_/CoFe_2_O_4_ is 1-2 orders higher than single-phase multiferroic materials. Later then, a massive research work was carried out on ME composites including both bulk and thin-film materials due to their great potential for multifunctional RF/microwave devices. In comparison with bulk materials, thin-film ME composites have some distinctive advantages such as low interface losses, complementary metal–oxide–semiconductor (CMOS)-compatible fabrication process, etc. Therefore, they are more promising candidates for integrated RF/microwave ME devices.

### 1.2. Magnetoelastic Materials

Owing to the renaissance of multiferroic materials, research works on magnetoelastic materials are increasing year by year as well. The year number of papers published on multiferroic, magnetoelectric and magnetoelastic is shown in Figure 1, the data of which is recorded from GOOGLE SCHOLAR. It is clear that the number of papers on multiferroic and magnetoelectric keeps rising since 21st century, which means researchers become more interested in this topic. The number of papers published on magnetoelastic also increases slowly in recent years result from the revival interest in ME effect. The selection of appropriate materials plays a key role in fabricating ME devices with good performance. The properties of different typical piezoelectric and magnetostrictive materials used for ME devices are listed in Table 2. Pb(Zr,Ti)O_3_ (PZT)-based ceramics are well-known piezoelectric materials that have been widely employed in ME devices due to their high piezoelectric coefficients and low cost. For the magnetostrictive phase, Galfenol (FeGa) and FeCoSiB with large magnetostriction constants and piezomagnetic coefficients have been mostly used in thin-film ME devices. There are a variety of properties for both piezoelectric and magnetic phases that need to be considered when developing novel ME applications, such as dielectric loss, polarization, piezoelectric constant, electromechanical coupling factor, magnetization, coercive magnetic field, magnetostriction, etc.

Magnetoelastic effects refer to the couplings between magnetic and elastic properties of a material, which can be divided into two categories: direct and inverse effects. The best-known direct magnetoelastic effects are magnetostriction and ΔE effects while some of the inverse effects are described by special terms such as Villari effect and Matteuci effect in the literature. The detailed description of magnetoelastic effects are listed in Table 3. Magnetostrictive materials have been playing an important role in applications ranging from actuators, sensors and energy harvesters. Here, we focus on the magnetostriction of magnetoelastic effects. By definition, magnetostriction means the change of shape or dimensions of a material during the process of magnetization. The magnetostriction was first identified by James Joule in 1842 when observing a sample of iron [64]. It can be categorized into spontaneous and forced magnetostriction, which are called volume and Joule magnetostriction. The illustration of volume and Joule magnetostriction of a spherical sample is shown in Figure 2. The isotropic expansion of the sample near Curie temperature under zero magnetic field is due to the spontaneous magnetostriction, while the Joule magnetostriction results in the elongation of the sample in the direction of magnetic field. The deformation in the orthogonal direction to the field with the opposite sign and half the amplitude is called transverse magnetostriction. There are many other factors that can lead to Joule magnetostriction such as the stress, volume change at high magnetic fields, etc. which are out of the scope of this review. Figure 3 shows the magnetostriction curves for volume and Joule magnetostriction. As shown in Figure 3a, a high volume magnetostriction of 0.82% in FeRh was discovered above room temperature by Ibarra and Algarabel [65], which experimentally demonstrated the predicted metastable ferromagnetic high-volume state within the antiferromagnetic phase by applying magnetic field. Dong et al. [66] reported the Joule magnetostriction values of some typical magnetostrictive thin films including Ni, Co, FeGaB, and FeCoSiB as shown in Figure 3b.

### 1.3. Magnetoelectric Applications

Due to the advantages of thin-film devices such as lightweight, low cost, high spatial resolution and CMOS-compatible fabrication process, thin-film ME heterostructures are preferred in specific applications where miniaturization of the device is crucial. Therefore, we focus on the recent progress on thin-film magnetostrictive materials in this article. Several types of integrated multiferroic devices based on thin-film ME heterostructures including ME sensors [18,63,67,68,69,70,71], inductors [35], filters [32] and antennas [43,72,73] are presented in this section. Based on the direct ME effect, a giant ME coefficient (α_ME_) as high as 5kV/cm Oe was obtained by the direct deposition of FeCoSiB on Si substrate with a Si/SiO_2_/Pt/ AlN/FeCoSiB layer stack [71]. The high quality of AlN and FeCoSiB films allows the enhancement of limit of detection (LoD) to be an extremely low value of 400 $fT/Hz^{1/2}$ at the mechanical resonance frequency of 867 Hz. Here, LoD is also frequently referred to as equivalent magnetic noise floor and given in the equation: $\mathrm{LoD}=\sqrt{\frac{P}{S^{2}}}$, where P is the power spectral density of the arbitrary quantity in units of au^2^/Hz, such as V^2^/Hz and rad^2^/Hz; S is the magnetic field sensitivity of the arbitrary quantity in units of au/T, such as V/T and rad/T. Therefore, the unit for LoD is $T/\sqrt{Hz}$. The single cantilevers were cut from wafers with a width of 2.2 mm and a length of 25.2 mm as shown in Figure 4a. These cantilevers were mounted on printed circuit boards (PCBs), and the top-bottom electrode connections were established manually. Operating at the working point with a bias dc field of $2.1\times 10^{-4}T$ and a constant ac driving field of $1\times 10^{-7}T$, the ME coefficients of the sensor with respect to frequency were measured and pictured in Figure 4b. The electromechanical resonance frequency and quality factor fitted to Lorentzian equation were observed as 867 Hz and 310. By performing 15 consecutive sensitivity measurements and averaging them, an estimation of the LoD of the sensor was determined as $400\pm 37\;fT/Hz^{1/2}$ and shown in Figure 4c. The influences of different noise regimes on the LOD of a ME sensor were investigated by measuring LoD at various frequencies, which is depicted in Figure 4d. Based on the converse ME effect, a voltage tunable inductor and bandpass filter were successfully demonstrated by Lin and Gao et al. [32,35] The schematics and SEM images are presented in Figure 5. The solenoid structure using FeGaB/Al_2_O_3_ multilayer was bonded to a lead magnesium niobate-lead titanate (PMN-PT) piezoelectric slab to fabricate the tunable inductor. Figure 5b shows the measured inductance under different E-field applied across the thickness of the PMN-PT slab, in which the inset shows the SEM picture of the inductor. A high tunable inductance of >100% is obtained over a large frequency range from 0 to 2 GHz and a peak inductance tunability of 191% at 1.5 GHz is obtained. The quality factor is also enhanced more than 100% over the frequency range from 0 to 1.5 GHz. Two coupled elliptic-shape nano-mechanical resonators consisting of FeGaB/Al_2_O_3_ multilayer and AlN film are excited in in-plane contour mode to realize the electromagnetic (EM) transduction. The dependence of the measured resonant frequency on the applied magnetic field was measured and shown in Figure 5d. Both E-field and H-field tunability were also measured and the frequency dependence on DC voltage across the thickness of AlN film is shown in ref. [13]. The H-field frequency tunability of 5 kHz/Oe and E-field frequency tunability of 2.3 kHz/Oe were observed in same device.

Utilizing both direct and converse ME effects, the unprecedented demonstration of ultra-compact ME NEMS antennas was proposed by us in 2017 [43] and was expected to have great impacts on our future antennas and communication systems. Two structures were designed to demonstrate the novel receiving and transmitting mechanism: nano-plate resonator (NPR) and thin-film bulk acoustic resonator (FBAR). The mechanism of the new antenna is explained as follow: from the receiving aspect, the magnetostrictive layer can detect the RF magnetic field component of the EM wave and induce an acoustic wave on the ferromagnetic layer. When this acoustic wave transfer to the piezoelectric thin-film, the dynamic voltage/charge would be generated due to the direct piezoelectric coupling; Reciprocally, from the transmitting aspect, the mechanical resonance, which is generated by applying RF electric field to the MEMS resonator, would induce acoustic wave that can be directly transfer to the upper ferromagnetic thin film. Then a dynamic change of the magnetization induced by acoustic wave due to the strong magnetostriction constant would generate magnetic current for radiation. As shown in Figure 6a,b, the antenna measurement setup is consisted of a horn antenna and our ME antenna whose radiative element is the suspended FeGaB/AlN circular disk. The electromechanical resonance frequency (f_r,FBAR_) of the ME FBAR is defined by the thickness of the circular resonating disk and can be expressed by $f_{r,\;FBAR}\sim \frac{1}{2T}\sqrt{\frac{E}{\rho }}$, where T is the total thickness, E is the effective Young’s modulus, and ρ is the effective density of the ME disk. The performance of ME FBAR was measured and presented in Figure 6c,d,a resonance frequency of 2.53 GHz and calculated gain of −18 dBi were achieved from the results of return loss (S_11_), transmission and receiving (S_12_ and S_21_) curves. A non-magnetic aluminum (Al) FBAR structure was fabricated as a control device to demonstrate that the source of radiation is from the ME coupling. A similar electromechanical properties of the control device is shown in Figure 6e, however, the much lower gain in Figure 6f suggests that the ME coupling dominates the radiation of FBAR antennas.

## 2. Fabrication and Characterization Methods

### 2.1. Fabrication Techniques of Thin Films

Due to the advantages of thin films, there are a variety of thin-film preparation techniques that have been developed such as physical vapor deposition (PVD) [74,75], chemical vapor deposition (CVD) [76,77], atomic layer deposition (ALD) [78,79], and sol-gel [80] etc. Here, we focus on the description of PVD process which is the most commonly used method for thin film deposition. PVD is a thin-film deposition process in a vacuum and plasma environment in which films with thickness in the range of a few nanometers to a few micrometers are produced. The source materials are vaporized and nucleated onto a substrate surface in a vacuum environment to fabricate the thin films. The vapor phase is usually consisted of plasma or ions. There are three major PVD process: evaporation, ion plating and sputtering, where the mechanisms for creating vapor phase are thermal effect, ion bombardment and gaseous ions, respectively. Tungsten wire coils are normally utilized as the source for heating the target materials and the deposition rate of the thermal evaporation is high compared to other PVD processes. Different configurations of evaporation such as molecular beam epitaxy (MBE) and activated reactive evaporation (ARE) are exploited to improve the films’ qualities as well. Based on the sources of sputtering deposition, various sputtering processes were developed such as diode sputtering, reactive sputtering and magnetron sputtering etc. Our PVD system is magnetron sputtering which is able to do both DC and radio frequency (RF) sputtering. DC voltage is provided between the anode and cathode, which are used to place substrate and target material, to sustain the glow discharge. The target material for DC sputtering is usually metallic so that the glow discharge can be maintained. RF sputtering is specifically designed to avoid charge building up on the surface of target materials, such as insulators. The magnetic field is exploited to restrict the movement of the secondary electrons emitted from the target surface and therefore efficiently increase the deposition rate.

### 2.2. Characterization Methods of Thin Films

The properties of magnetic and piezoelectric phases are crucial on achieving large ME coupling strength, in which piezomagnetic and piezoelectric coefficients play significant roles among a variety of thin film properties. Given that ME heterostructures are widely applied in RF devices, extensive works have been carried out to develop the characterization methods for thin film materials. Some well-known techniques have been developed for decades, such as X-ray photoelectron spectroscopy (XPS) and X-ray diffraction (XRD), which are utilized to measure the compositions and crystal structures of thin films. Magnetic properties including hysteresis loops and ferromagnetic resonance (FMR) curves can be measured using vibrating sample magnetometers (VSM), superconducting quantum interference devices (SQUID), and FMR tester. A waveguide and vector network analyzer (VNA) are utilized to extract the permeability, loss tangent and FMR linewidth of magnetic thin films from the measured S parameters [81]. With regard to characterization of magnetostriction constants, both direct and indirect methods were proposed with different mechanisms [82,83,84,85,86,87]. Since the deflection of a cantilever structure with the magnetic film on the upper side is proportional to the magnetostriction, an instrument was devised to measure the magnetostriction as a function of applied field by measuring the change of mechanical resonance frequency [83]. By utilizing a continuous laser beam system, a non-contact method was developed to measure the saturation magnetostriction of a thin soft-magnetic film deposited on a thin glass substrate under a rotating magnetic field [84].

In addition to the increased interest in magnetostriction, ∆E effect that represents the change of Young’s modulus as a function of magnetic field has drawn a great amount of attention to build ultra-sensitive ME sensors. Based on the non-contact optical technique, a simple, compact, and sensitive system to measure the magnetostriction and ∆E effect of magnetic thin films was developed by us recently [66]. The schematic of the proposed magnetostriction tester is shown in Figure 7a. A rotating adjustable AC magnetic field with a maximum value of 300 Oe is generated by two pairs of mutually perpendicular set Helmholtz coils with equivalent amplitude and 90° phase shift current. Similar to the cantilever method for measuring piezoelectric coefficients, the tip displacement of the cantilever is detected by a MTI-2000 fiber-optic sensor due to the strain change in the magnetostriction film. Figure 7b shows the schematic of the proposed delta-E tester, which is very similar to the magnetostriction tester. Second vibration mode of the cantilever is excited by applying a 0.1 Oe AC driving field through a solenoid with the frequency swept around 1555 Hz. A DC bias magnetic field is generated by the Helmholtz coil to induce magnetic domain change in the magnetostrictive films, and thus resulting in the change of Young’s modulus. According to the modified Euler-Bernoulli beam theorem, the change of Young’s modulus can be derived by measuring the shift in resonant frequency of the cantilever. Some typical magnetostrictive films such as Ni, Co, FeGa etc. with well-known magnetostriction values are used to calibrate the system. Figure 7c shows the measured magnetostriction and piezomagnetic coefficient for as-deposited and annealed FeGaB thin films. The change in Young’s modulus of FeGaB thin film is also measured and shown in Figure 7d by using delta-E tester.

Both the fundamental studies of magnetism and boosting of magnetic industry are greatly influenced by the imaging of magnetic microstructures. One of the most effective ways to characterize the performance of ME-based devices is the analyzation of magnetic domain activity. There are only a few methods for observing magnetic domains down to device level including magnetic force microscopy (MFM) [88], scanning electron microscopy with polarization analysis (SEMPA) [89] and soft X-ray magnetic circular dichroism–photoemission electron microscopy (XMCD-PEEM) [90]. Due to magnetic fields emitting from the domain in the magnetic materials, the force gradient acting on a small magnetic tip can be sensed by using ac detection method [88]. A high contrast and spatial resolution of magnetic domain was seen from the magnetization patterns. SEMPA is another imaging technique with high spatial resolution for surface magnetic microstructures on the 10-nm-length scale [89]. By employing a SEM electron optical column, the secondary electrons whose spin polarization are directly related to the local magnetization orientation are excited and emitted from the ferromagnetic sample surface. Therefore, the surface magnetization map can be generated when the spin polarization of secondary electrons is analyzed across the sample surface. A very large average velocity of 600 m/s was detected for domain-wall motion in the NiFe layer by using XMCD-PEEM [90]. In particular, magnetic domain imaging by magneto-optical Kerr effect (MOKE) microscopy offers direct access to the behavior of local magnetization [91,92,93,94]. MOKE microscopy based on the Kerr and the Faraday effects is one of the most prominent techniques for observing magnetic domain and is able to visualize the magnetic dynamics on fast time-scales. Allowing for the direct imaging under a continuous field excitation, a picosecond wide-field MOKE microscopy for imaging magnetization dynamics is demonstrated in [92]. An example of nanosecond domain wall displacements and spin-wave generation are observed and displayed in Figure 8 [93]. Figure 8a shows the precessional domain wall motion with out-of-plane magnetization components of a (Fe_90_Co_10_)_78_Si_12_B_10_ sample. The spin waves, which origin from the domain wall precession within the central magnetic domains shown in Figure 8b, propagate with a velocity of 650 m/s. Moreover, the direct imaging of standing spin waves at field excitations in a Co_40_Fe_40_B_20_/Ru/ Co_40_Fe_40_B_20_ anti-dot array at 2 GHz is exhibited in Figure 8c.

## 3. Magnetostrictive Thin Films

### 3.1. Rare-Earth-Transition Metal Inter-Metallics

Magnetostriction was first reported by James Joule in the early 1840s, who observed the change of length in iron particles when exposed to magnetic fields [64]. Villari discovered that applying stress to magnetostrictive materials changes their magnetization [95]. However, magnetostrictive materials have not been extensively used until World War II, when nickel-based alloys were employed in building sonar applications. The magnetostriction of 3d transition metals Fe, Co and Ni can produce very high forces, but the range of motion is small. The extraordinary magnetostrictive properties of the rare-earth elements such as terbium and dysprosium were first recognized in the early 1960’s, which can achieve magnetostriction of up to 10000 ppm at low temperatures [96]. However, they do not demonstrate significant magnetostriction at room temperature where practical devices must operate due to the low Curie temperature. The solution to this problem was pointed out by Callen in 1969, who suggested there was great promise that strong magnetism of the transition metals such as iron, which exhibit higher Curie temperatures, could increase the rare-earth magnetic order at high temperatures [97]. In attempt to produce large magnetostriction at room temperature, rare-earth elements were alloyed with 3d transition elements as rare-earth-transition metal inter-metallics.

#### 3.1.1. TbDyFe

In 1979, the magnetic materials group at Carderock which was funded by the Office of Naval Research (ONR) discovered that the magnetocrystalline anisotropy of rare-earth-transition metal inter-metallics can be significantly reduced by adding Tb and Dy in certain proportion into iron. This led to the development of the Terfenol-D alloy (Tb_0.3_Dy_0.7_Fe_2_), which exhibits giant magnetostriction of about 2000 ppm at room temperature [98]. The large strain produced by Terfenol-D is useful in applications as production of high amplitude, low frequency sound wavs in water, certain types of strain gages, vibration compensation and compensation for temperature induced strains in large laser mirrors. One of the draw backs of Terfenol-D is its brittleness in withstanding tensile stress, which limits its ability to withstand shock loads or operate in tension.

In terms of Terfenol-D thin films, several efforts on the fabrication techniques and characterization studies have been conducted during the last two decades [98,99]. Terfenol-D thin films are commonly prepared by magnetron sputtering with a single alloy target or by co-sputtering with multiple targets. However, the as-deposited films are always amorphous and their magnetoelastic behaviors and magnetic properties are inferior than their bulk counterpart. To overcome these drawbacks and enhance their mechanical and magnetic properties, the thin films must be crystallized either by post-annealing or by depositing the film on a heating substrate. In 1994, Williams P. I. et. al. observed in-plane magnetostriction coefficient of 500 ppm in Tb_0.3_Dy_0.7_Fe_2_ polycrystalline thin film as shown in Figure 9 [100]. However, the saturation field achieved at 4 kOe was still too high, which made it impractical for the usage in MEMS applications, as it would be very difficult to apply such a high bias magnetic field on chip.

#### 3.1.2. SmFe and SmFeB

In addition to Terfenol-D alloys, samarium-based alloys have also been of great research interests due to their high Curie temperature. Unlike Terfenol-D alloys, which start to degrade their magnetostrictive performance at 200 °C, Samarium-based alloys can be applied to much higher temperature environment without losing their magnetic properties [101]. One special character of Sm-Fe alloy is their negative sign of giant magnetostriction, which lies in the relationship between the magnetoelastic behaviors and the stress state of the magnetostrictive materials. Specifically, the compressive stress can increase the strain for magnetostrictive materials with positive sign, whereas decrease the strain for materials with negative sign. Therefore, in device applications using bulk magnetostrictive materials, usually they are intentionally applied with compressive stress rather than tensile stress for convenience to achieve the largest strain. However, in thin film applications, it is not necessary to apply any stress on the thin films, therefore, Sm-Fe based alloys do not suffer from the problem related to the reduction of strain due to the applied stress on negative magnetostrictive materials. Another advantage is that the light rare-earth element samarium is more abundant than the heavy rare-earth element terbium. Therefore, Sm-Fe based alloys have a price advantage over Terfenol-D alloys. These advantages lead to the investigation and development of Sm-Fe based thin films with good magnetostrictive characteristics.

In 1998, Kim et al. conducted a systematic investigation on the magnetostrictive behavior of Sm-Fe and Sm-Fe-B thin films over a wide composition range from 14.1 to 71.7 at% Sm [102]. As displayed in Figure 10, good magnetostrictive characteristics has been found at low magnetic fields in both Sm-Fe and Sm-Fe-B thin films. At 100 Oe, magnetostriction of −350 ppm and −470 ppm have been achieved for Sm-Fe with 25% Sm and for Sm-Fe-B with 27–31% Sm respectively. The addition of B has formed an amorphous phase in Sm-Fe-B thin films, which effectively modified the microstructure and thereby improved the magnetic properties. At the same time, a very high $\lambda_{s}$ of −1200 ppm has been achieved with an optimum Sm content at 36.8% for Sm-Fe-B thin films, which is three times larger than Tb-Fe based thin films. The excellent magnetostrictive characters of Sm-Fe and Sm-Fe-B thin films made them good candidates for the applications in MEMS device.

### 3.2. Rear-Earth-Free Alloys

The well-known problem of the magnetostrictive rear-earth-transition metal films is that a large bias magnetic field is usually required to achieve giant magnetostriction due to the large magnetocrystalline anisotropy induced from the rear-earth elements, which is particularly impractical for these magnetostrictive thin films to be applied on MEMS devices. Also, at RF and microwave frequencies, even small amount of rear-earth elements can result in a very lossy magnetic film. Recently, strong magnetostrictive behaviors have been observed in Fe-Ga and Fe-Al alloys, which exhibit large magnetostriction of more than 400 ppm for Fe_81.3_Ga_18.7_ and 200 ppm for Fe_83.4_Al_16.6_ respectively [103,104]. In these alloys, it has been found that the giant magnetostriction always occurred in the vicinity of ordered D03 and disordered A2 phase boundaries. In addition, since the first report on the magnetostrictive properties of Fe-Co alloy by Masiyama, much efforts have also been devoted in developing giant magnetostrictive Fe-Co alloys [105].

#### 3.2.1. FeGa Based Alloy

To overcome the intrinsic drawbacks in RFe alloys, researchers began to search for other rear-earth-free magnetic materials with large magnetostriction, low saturation field, low cost and high mechanical durability. In 1999, the group at Carderock which was funded by the Office of Naval Research (ONR) invented a new exciting material, the Fe-Ga (Galfenol) alloy system. Galfenol alloys exhibit large saturation magnetostriction of 400 ppm in single crystal form and 300 ppm in polycrystalline form with a much lower saturation field of less than 200 Oe [106,107]. In addition, these alloys are stable over a wide temperature range from −265 °C to 150 °C and can sustain approximately 350 MPa tensile stress. Galfenol alloys can be stress-annealed by applying a compressive force during the annealing process, which eliminates the need to apply external compressive stress for certain usage [108].

Although single crystal Fe-Ga alloys exhibit the advantage of large magnetostriction at relatively low saturation field, and can tolerate substantial compressive and tensile stress, however, they have large ferromagnetic resonance linewidth of 400–600 Oe at X band, which make them very lossy for integration into microwave devices [109]. Previous studies have found that the addition of trace amounts of metalloid atoms of C, B, and N into binary alloys has a beneficial effect on increasing magnetostriction. In 2007, Lou et al. reported magnetostrictive behavior of Fe-Ga-B thin films with different B contents [62]. As shown in Figure 11, through the incorporation of boron at 9–12 %, the gain size has been refined and the magnetocrystalline anisotropy has been effectively reduced, which results in the combination of excellent magnetic softness with a maximum saturation magnetostriction of 70 ppm at saturation field lower than 20 Oe. The reason for this, is that small amount of B atoms would stay in the interstitial sites of the bcc lattice and destabilize the D03 phase which is detrimental to magnetostriction, while higher amount of metalloid atoms would form clusters that decrease the magnetostriction constant [110]. It is also believed that a low concentration of metalloid atoms could lead to atomic pairs such as Ga–Ga or B–B, which enhance the magnetostrictive behavior.

Beside boride Fe-Ga alloys, the magnetostrictive behavior of carbide Fe-Ga alloys have also been investigated. In 2018, Liang et al., reported the C-concentration dependence on magnetostriction of Fe-Ga-C films [61]. Figure 12 shows the saturation magnetostriction constants and maximum piezomagnetic coefficients of Fe-Ga-C films with different C content. The saturation magnetostriction $\lambda_{S}$ reaches the maximum value of 81.2 ppm when C atoms incorporate into a Fe-Ga lattice with 11.1 at. % C content and continually reduces to 22 ppm with C composition of 14.2 at. %. The gradually increasing value of $\lambda_{S}$ with the C content increasing from 0 at. % to 7.1 at. % is illustrated by the slowly transition from crystalline structures to amorphous state of Fe-Ga-C films.

When the C content becomes more and more, the $\lambda_{S}$ reaches its maximum value near the amorphous/crystalline phase boundary, where Fe-Ga-C films change from crystalline structure to amorphous phase. The maximum $\lambda_{S}$ of Fe-Ga alloy systems occurs at the composition of Fe_80_Ga_20_, which is near the A2/D03 phase boundary and is consistent with Fe-Ga-C systems. The maximum $\lambda_{S}$ value of Fe-Ga-C films is 81.2 ppm and is six times larger than that of the binary Fe-Ga film, while the maximum $\lambda_{S}$ value of Fe-Ga-B systems only tripled, which means a better influence of C addition on enhancing magnetostriction constant of Fe-Ga films. Moreover, Fe-Ga-C films are more magnetic sensitive than that of Fe-Ga-B films due to their higher piezomagnetic coefficient of 9.71 ppm/Oe.

#### 3.2.2. FeCo Based Alloys

It is well-known that Fe–Co alloy systems have very soft magnetic properties. Besides, Fe-Co alloys own the largest saturation magnetic flux density among the Fe, Co and Ni systems due to its largest number of Bohr magnetons according to the Slater–Pauling curve [111].

It is reported that the magnetostriction of Fe–Co binary alloys greatly rely on their composition over a wide range in Masiyama’s researches. There were two obvious peaks of magnetostriction that were clearly measured, the 1st peak being around the composition of Fe_30_Co_70_ (90 ppm) and the other one being around the composition of Fe_60_Co_40_ (70 ppm) [105]. From the phase diagram of the Fe-Co binary systems, we can see a phase boundary between the bcc structure and its coexisting region with fcc phase at a Co content of ~75 at. %. It was found in the previous report that the magnetostriction of Fe-Co systems could be enhanced by forming a phase boundary, which is analogous to the magnetostriction augment of Fe-Ga alloys near the D03/A2 phase boundary. And they indeed achieved the largest magnetostriction at a Co content of ~70 at. % in that report.

In 2019, Wang investigated the magnetostrictive behavior of (Co_0.5_Fe_0.5_)_x_C_1-x_ alloy thin films with different carbon content [112]. Figure 13 presents the magnetostriction results of these Co-Fe-C films with increasing carbon content as a function of the in-plane driving magnetic field. There is no magnetostrictive response for those samples with carbon content lower than 4% when the magnetic field is less than 30 Oe. However, when the carbon content of Co-Fe-C films is higher than 4%, the AC driving field reduces to 20 Oe and gradually becomes smaller. This magnetostrictive behavior indicates that the carbon content of 4% is a critical value for the Co-Fe-C films’ phase boundary between nanocrystalline and amorphous structures. As shown in Figure 13b, black squares denote the magnetostriction constants of Co-Fe-C films. The saturation magnetostriction $\lambda_{S}$ reaches a maximum value of 75ppm when carbon atoms are added to a Fe-Co lattice with 5.2 at. % carbon content, and then gradually reduces to 10 ppm at carbon content of 15.8 at. %.

The piezomagnetic coefficient of the Co-Fe-C films was shown by the red cycles in Figure 13b. The curve can be divided into three regions: When the carbon content is less than 4 at. %, the Co-Fe-C films remain to be bcc nanocrystalline phase; as the carbon content increases and is less than 6 at. %, there is a coexistence region of both nanocrystalline and amorphous phase and the phase boundaries are labelled as the dotted orange and green lines; when the carbon content goes beyond 6 at. %, they transition to amorphous structure. A clear peak is obtained in the coexistence region and the largest value of piezomagnetic coefficient is achieved of 10.3 ppm/Oe at 4.8 at. %. They also confirmed the coexistence region by analyzing FFT results of the HRTEM images, which are shown in Figure 14. The formation of atom pairs or clusters during the doping of metalloid atoms also destabilize the D03 phase. Therefore, the magnetostriction constants and piezomagnetic coefficients of Co-Fe-C films are optimized with the carbon content ranging from 4% to 6%. Compared to the reported value of magnetostriction constants for Ga-Fe-B and the widely used Co-Fe-B films, the values measured in this paper are much higher. Moreover, the maximum piezomagnetic coefficient dλ/dH of Co-Fe-C films is much larger than other well-known magnetostrictive materials such as Metglas and Terfenol-D.

### 3.3. Exchange Coupling in Multilayers

In addition to the state-of-the-art magnetostrictive single layer materials, the combination of different materials in multilayer structure has recently been proposed as an alternative approach to develop giant magnetostrictive thin films with low loss and low saturation field. In multilayer thin films, the magnetic properties in different layers can be synthesized through the process of exchange coupling, which arises when the thickness in each layer is smaller than the length of magnetic exchange [113]. Through exchange coupling, spin information can be transmitted between two different magnetic materials, and the overall magnetic properties are determined by the average value of each layer. By combining two different materials with excellent softness and giant magnetostriction, exchange coupling can be used to reduce the saturation field and improve the piezomagnetic coefficient.

#### 3.3.1. FeGa/NiFe Multilayers

In 2019, Shi et al. studied the effect of exchange coupling between Fe_83_Ga_17_ and Fe_50_Ni_50_ multilayer films for enhancement of the ME effect [114]. As shown in Figure 15, all of the magnetic domain structures are stripe domain. However, with the total film thickness fixed at 480 nm, as the thickness in each layer of Fe-Ga and Fe-Ni become thinner, the stripe width become narrower. This means the domain density has been increased, which intensified the domain rotation and domain wall motion, thereby enlarged the magnetostriction. It is believed that stripe domain structure indicates the films have weak out-of-plane anisotropy [115], and the magnetic moments on the surface of the films rotate upward and downward alternatively. With the Fe-Ni layers, the permeability increased and the total magnetocrystalline anisotropy reduced, which in turn make the 90-degree domain wall motion easier. For the (8 nm Fe-Ga/8 nm Fe-Ni)_30_ multilayers, the small magnetic anisotropy resulting from the exchange coupling makes the domains rotate more easily under low magnetic field. The rotation of the magnetic domains in the Fe-Ni layers also drive the rotation of the magnetic domains in the Fe-Ga layers, which makes the Fe-Ni layers act like a lubricant. As shown in Figure 16b, the d_33,m_ of the (8 nm Fe-Ga/8 nm Fe-Ni)_30_ multilayers achieved to 8.1 ppm/Oe. Compared to the Fe-Ga-B film, the piezomagnetic coefficient of the multilayer film has an increment of about 14%.

#### 3.3.2. TbFe/FeCo Multilayers

In 1999, Quandt et at. reported exchange coupled Tb_40_Fe_60_/Fe_50_Co_50_ multilayers which exhibits high saturation magnetoelastic coupling coefficients in combination with low saturation field of 20 mT [116,117]. It has been found that the saturation magnetostriction was strongly depended on the layer thickness ratio and the post annealing treatment, as shown in Figure 17c,d. 

The optimum data of the TbFe/FeCo multilayers show that post annealing at 250°C improves the magnetoelastic behavior, and the coercivity has been reduced from 10 mT to 2 mT, which reflects that post annealing can release almost all stress induced anisotropies, leading to a stress-free state of these multilayers. It is also believed that under these conditions any significant diffusion does not exist and nanocrystallization begin to appear in the TbFe layers, which leads to the enhancement of the saturation magnetoelastic coupling coefficient by 15%. At the same time, the overall temperature behavior is also compromised by the exchange coupled multilayers, and it is expected that the strong Fe_50_Co_50_ moments can help stabilized the Tb_40_Fe_60_ moments, resulting in an increment of the Curie temperature.

### 3.4. Heat Treatment on Magnetostrictive Thin Films

#### 3.4.1. Stress Release by Post Annealing

As a highly magnetostrictive material, FeCoSiB is one of the most commonly used composition in Metglas foils. In 2002, Quandt et al. investigated the optimization of the ΔE effect and the magnetostriction of amorphous (Fe_90_Co_10_)_78_Si_12_B_10_ thin films by magnetic field annealing [118]. They discovered a strong correlation between the magnetostrictive susceptibility and the stress state, which can be controlled by the annealing temperature. As shown in Figure 18c, the as-deposited FeCoSiB film was initially in compressive stress state before the annealing temperature reach to 170 °C. However, as the annealing temperature continue to increase, the stress state gradually switched to tensile stress. After annealing treatment, the compressive stress in the film has been effectively reduced, leading to the reduction of anisotropy energy. The change of the Young’s modulus with applied magnetic field has been shown in Figure 18d. After magnetic annealing, a distinguished Young’s modulus reduction of 50 GPa has been observed, which contributed to 30% change of the Young’s modulus at saturation state, assuming the Young’s modulus of FeCoSiB thin film at saturation is 150 GPa. In Figure 18a,b, we can see the magnitude and the field dependency of the magnetostrictive susceptibility are highly depends on the stress state and the corresponding annealing temperature. To achieve high magnetostriction at low magnetic fields, a precise control of film stress has to be executed. For positive magnetostrictive thin films, usually tensile stress state is favorable to obtain the optimum performance.

#### 3.4.2. Phase Boundary Changes by Cooling Processes

In 2011, Hunter et al. investigated the magnetostrictive behavior and microstructural properties of Co-Fe alloy thin films in relation to the composition and thermal process [119]. Strong dependence of magnetostriction on the cooling process has been found. The analysis of the microstructure through synchrotron X-ray micro-diffraction shows that the maximum magnetostriction occurs at the (fcc+bcc)/bcc phase boundary. As shown in Figure 19a,b, after annealing, the peak of magnetostriction shifts to lower Co composition by the same amount as the (fcc + bcc)/bcc phase boundary shifts, indicating that the peak magnetostriction is linked to this phase interface. At the temperature/composition close to the (fcc + bcc)/bcc phase boundary, a remarkable giant magnetostriction has been yield after annealing followed by quenching, which is more than three times of its as-deposited state.

The Co-Fe phase diagrams in Figure 19 b show that the bcc phase intersects with a mixed phase region of fcc Co and bcc Fe phases at temperatures lower than 912 °C and Co concentrations greater than 50%. The coexistence of the predominant bcc phase and a secondary fcc phase has also been confirmed by the TEM data shown in Figure 20c,d. It is believed that the precipitation of the fcc Co-rich grains into the bcc α-Fe matrix is the cause of the enhancement in magnetostriction, which function in the same way as the D03 precipitates in the Fe–Ga alloys.

## 4. Summary and Conclusions

Although lots of studies have been done on the magnetostrictive thin films, there are still many challenges and opportunities need to be addressed in the coming years. The essential figure of merits in developing thin film magnetoelastic materials are the piezomagnetic coefficient, coercivity, and FMR linewidth. Figure 21 summarizes the current status of the developed magnetostrictive materials in terms of these figure of merits. The direction of developing desired magnetostrictive materials goes to high saturation magnetostriction, high piezomagnetic coefficient with low coercivity and narrow FMR linewidth. The light rare-earth-based alloys, such as SmFe, tend to be the focus of researches in rare earth magnetostrictive materials due to the lack of heavy rare earth sources. As for rare-earth-free magnetostrictive materials, Fe-Ga and Co-Fe systems with large magnetostriction and low saturation field are found to be the most promising. New structures such as multilayers that combine properties of high magnetostriction and low anisotropy are promising for developing magnetoelastic materials with both softness and large magnetostriction constants. The stress state and the phase boundary can also be tuned to achieve the optimal magnetostrictive thin films through heat treatment process.

In summary, during last decades, extensive progress has been made for achieving high-performance magnetostrictive materials and different kinds of thin film based integrated multiferroic devices. In this review paper, we primarily devoted our efforts on the thin film types of magnetostrictive materials and their applications of RF MEMS devices based on strain-mediated ME coupling, which have drawn a great deal of interests in the past years. The performance of these RF MEMS devices was greatly benefited from the large piezomagnetic coefficients of the magnetostrictive materials, which leads to a strong ME coupling. From the science point of view, understanding, modelling and characterizing of thin film magnetostrictive materials are of great importance for the multiferroic society. From the materials point of view, new magnetostrictive materials with higher piezomagnetic coefficients are always more than welcome. From the perspective of engineering applications, integration and scale-up with existing silicon processing flows and electronics are in particularly valuable. Low-temperature process methods are desired for fabricating RF multiferroic devices with low loss as well. In addition, magnetic sensors have been introduced long ago to electrical and biomedical research areas and various types of magnetic field sensors are proposed [120]. Recently, flexible magnetic sensors that based on various mechanisms, such as anisotropic magnetoresistance (AMR) [121], giant magnetoimpedance (GMI) [122], etc. have received significant research interests because of their applications in wearable and portable devices. Magnetic field sensors that are fabricated on flexible substrates, which can be attached or transferred into the human body, grant us more freedom to design different healthcare devices. Therefore, flexible thin film magnetoelastic materials that can be compatible with wearable electronics are also significant for developing functional device applications. Through the better understanding in the origination of magnetostriction and the loss mechanism of magnetostrictive thin films, there exist a bright future and great challenges for thin film magnetoelastic materials. These materials will very likely facilitate the development of novel integrated RF multiferroic devices, which are compact, lightweight, and power efficient and will have huge influences on our daily life.

## Figures and Tables

**Figure 1 sensors-20-01532-f001:**
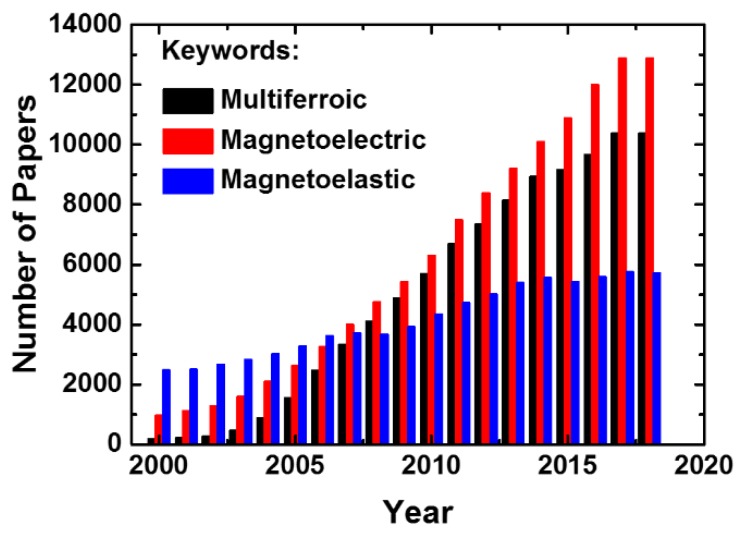
The year number of papers published on the multiferroic, magnetoelectric and magnetoelastic.

**Figure 2 sensors-20-01532-f002:**
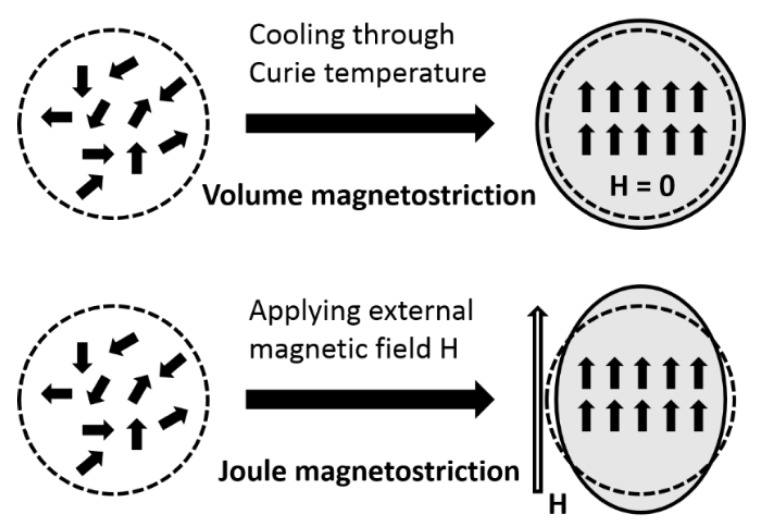
Illustration of volume and Joule magnetostriction of a spherical sample.

**Figure 3 sensors-20-01532-f003:**
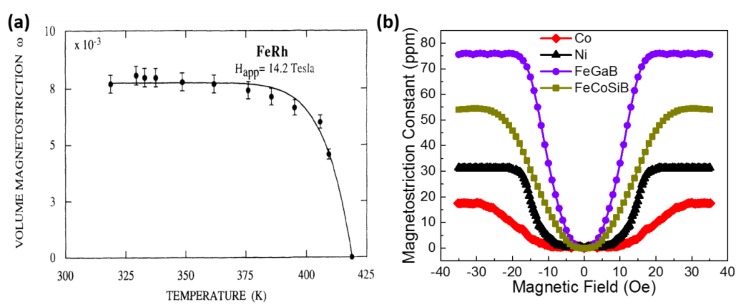
(**a**) Thermal dependence of the volume magnetostriction at 14.2 T for FeRh (the line is a visual guide). Reprinted with permission from [65]. Copyright [1994], American Physical Society. (**b**) Measurement results of the magnetostriction constant in the absolute value of different magnetic thin films (the line is a visual guide).

**Figure 4 sensors-20-01532-f004:**
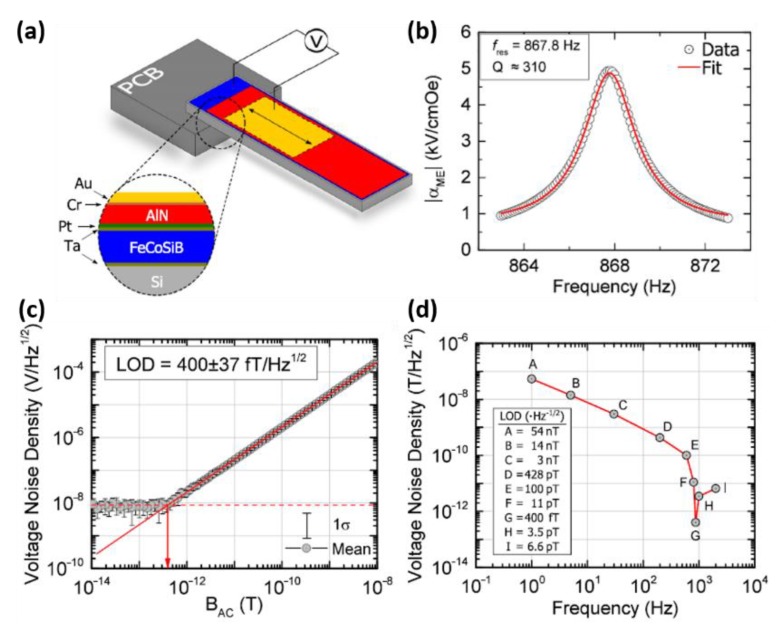
(**a**) Schematic illustration of the cantilever-based thin-film magnetic sensor using inversed bilayer ME heterostructures. (**b**) ME coefficient as a function of frequency ($H_{ac}=1\times 10^{-7}T$ and $H_{bias}=\pm 2.1\times 10^{-4}T$). The electromechanical resonance frequency was measured to be 867 Hz and the quality factor is determined as 310 from the applied Lorentzian fit. (**c**) Averaged LoD plot with error bars that indicate standard deviation. (**d**) The measured ME sensor’s LoD with respect to frequency. Reproduced with permission from [71]. Copyright [2016], American Institute of Physics.

**Figure 5 sensors-20-01532-f005:**
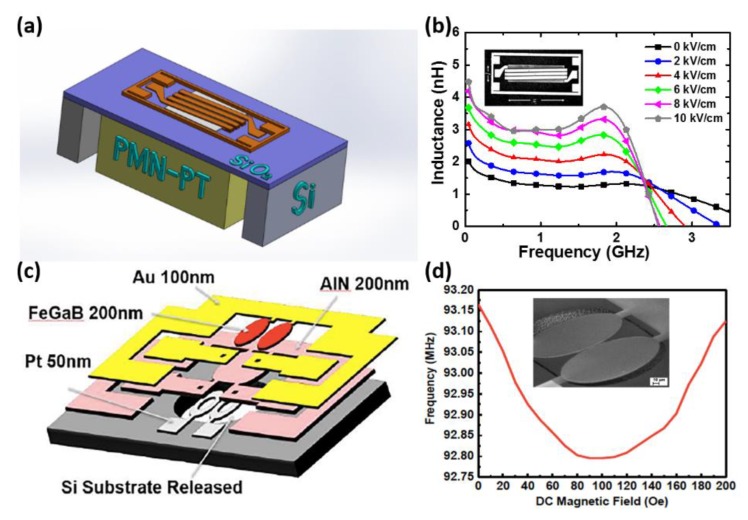
(**a**) Structure model of the voltage tunable inductor based on FeGaB/PMN-PT heterostructures. (**b**) Measured inductance under different E-field applied across the thickness of the PMN-PT slab (inset shows the SEM picture of the inductor). (**c**) Schematic of the voltage tunable filter based on FeGaB/AlN multilayers. (**d**) Dependence of the measured resonant frequency on the applied magnetic field (inset shows the SEM picture of the filter). Reproduced with permission from [32]. Copyright [2016], IEEE.

**Figure 6 sensors-20-01532-f006:**
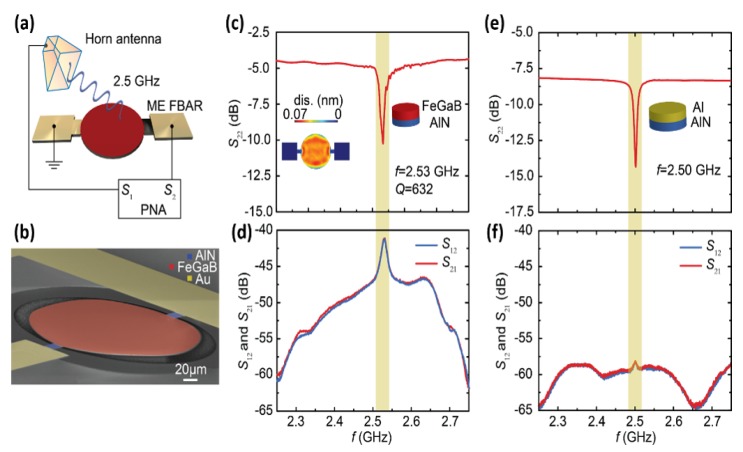
Thin-film based ME FBAR antennas. (**a**) Illustration of the antenna measurement setup. (**b**) SEM image of the fabricated the ME FBAR. (**c**) Return loss curve (S_22_) of ME FBAR. The inset shows the out-of-plane displacement of the circular disk at resonance peak position. (**d**) Transmission and receiving behavior (S_12_ and S_21_) of ME FBAR. (**e**) Return loss (S_22_) curve of the non-magnetic Al/AlN control FBAR. (**f**) Transmission and receiving behavior (S_12_ and S_21_) of the non-magnetic Al/AlN control FBAR. Reprinted with permission from [43]. Copyright [2017], Springer Nature.

**Figure 7 sensors-20-01532-f007:**
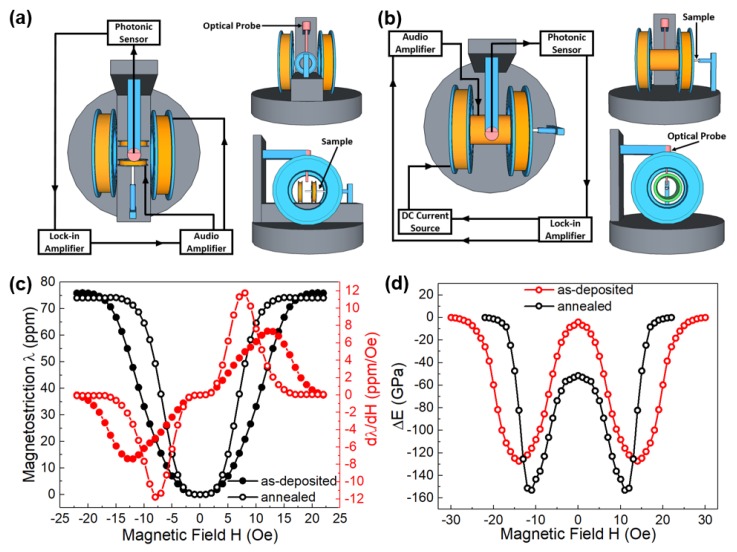
(**a**) Schematic of the proposed magnetostriction tester in three-view. (**b**) Schematic of the proposed delta-E tester in three-view. (**c**) Measured magnetostriction and piezomagnetic coefficient for as-deposited and annealed FeGaB thin films. (**d**) Measured delta-E effect for as-deposited and annealed FeGaB thin films. Reprinted with permission from [66]. Copyright [2018], American Institute of Physics.

**Figure 8 sensors-20-01532-f008:**
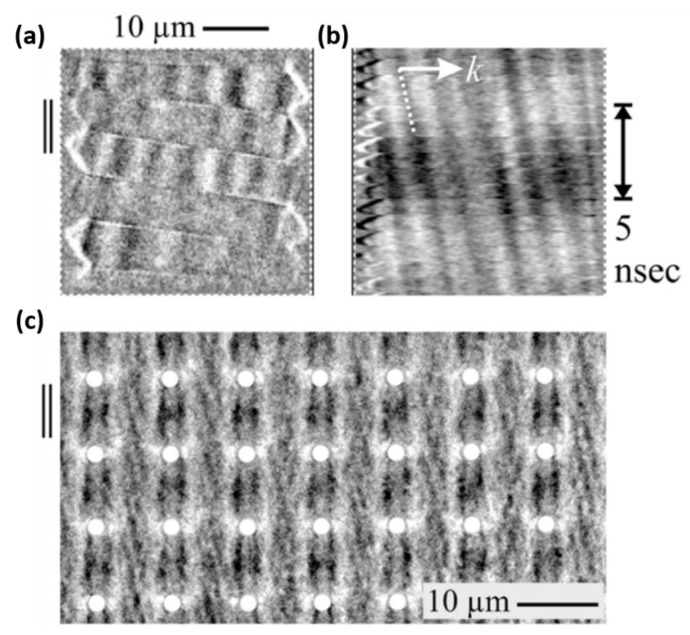
Domain images of magnetic dynamics for a magnetic element. (**a**) Domain wall motion at wall resonance. (**b**) Domain wall induced spin-wave propagation with time under a 100MHz field excitation. [sample: (Fe_90_Co_10_)_78_Si_12_B_10_(160 nm)] (**c**) Imaging of standing spin-wave modes in a Co_40_Fe_40_B_20_/Ru/ Co_40_Fe_40_B_20_ anti-dot array at 2 GHz. Reprinted with permission from [93]. Copyright [2015], American Physical Society.

**Figure 9 sensors-20-01532-f009:**
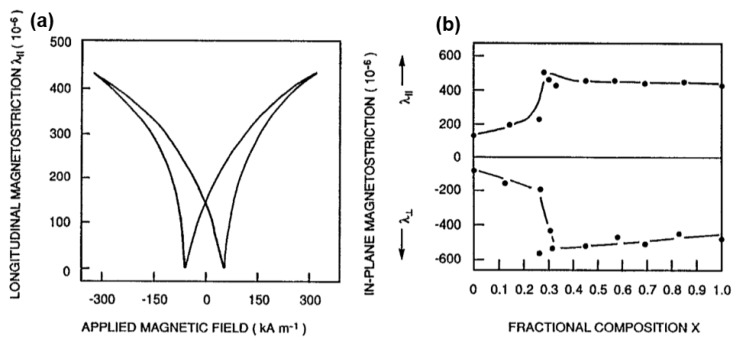
(**a**) In-plane, longitudinal magnetostriction λ_‖_ loop for a polycrystalline Tb_0.3_Dy_0.7_Fe_2_ film and (**b**) λ_‖_ and λ_⊥_ as a function of x for Tb_x_Dy_1-x_Fe_2_ films. Reprinted with permission from [100]. Copyright [1994], American Physical Society.

**Figure 10 sensors-20-01532-f010:**
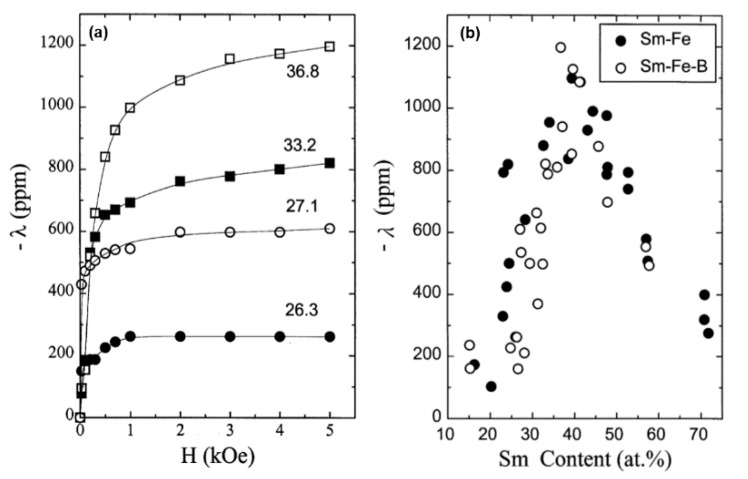
(**a**) The λ-H plots for some Sm-Fe-B thin films. The numbers at the curves indicate the Sm content in at% (**b**) The value of λ as a function of the Sm content at fixed magnetic fields of 5 kOe. The results of Sm-Fe thin films are indicated by filled circles while those of Sm-Fe-B thin films are denoted by open circles. Reprinted with permission from [102]. Copyright [1998], North Holland (Elsevier).

**Figure 11 sensors-20-01532-f011:**
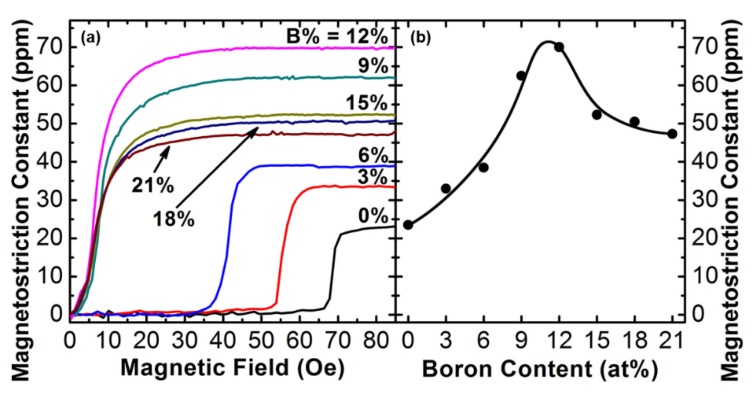
(**a**) Magnetostriction vs magnetic field. (**b**) Saturation magnetostriction constant vs B content. Reproduced with permission from [62]. Copyright [2007], American Institute of Physics.

**Figure 12 sensors-20-01532-f012:**
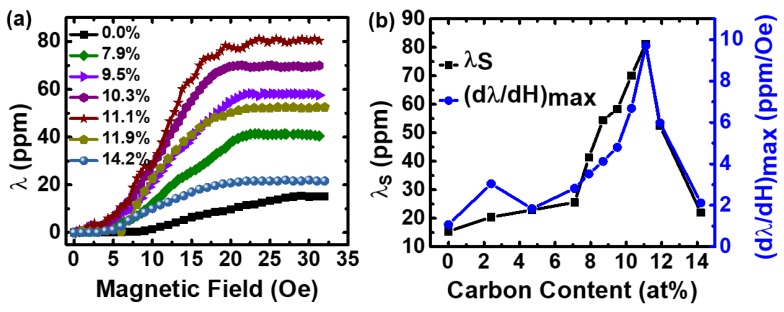
(**a**) Magnetostriction constant of Fe-Ga-C. (**b**) Saturation magnetostriction constant and maximum piezomagnetic coefficient of Fe-Ga-C. Reproduced with permission from [61]. Copyright [2019], IEEE.

**Figure 13 sensors-20-01532-f013:**
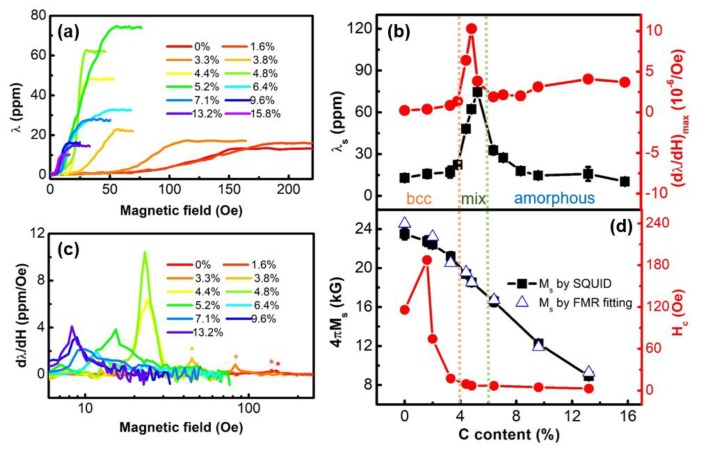
(**a**) Magnetostriction constant of (Co_0.5_Fe_0.5_)_x_C_1−x_ films with x = 0 to 15.8%. (**b**) The saturated magnetostriction constant (black) and piezomagnetic coefficient (red) with different carbon content. The structural boundaries of the alloy are marked with light orange and green dotted lines. (**c**) Piezomagnetic coefficient versus corresponding driving magnetic field for (Co_0.5_Fe_0.5_)_x_C_1−x_ films with x = 0 to 13.2%. (**d**) The M_s_ (black) and H_c_ (red) versus carbon content. The blue open triangles are the fitted M_s_ from broadband FMR measurement. Reproduced with permission from [112]. Copyright [2019], American Physical Society.

**Figure 14 sensors-20-01532-f014:**
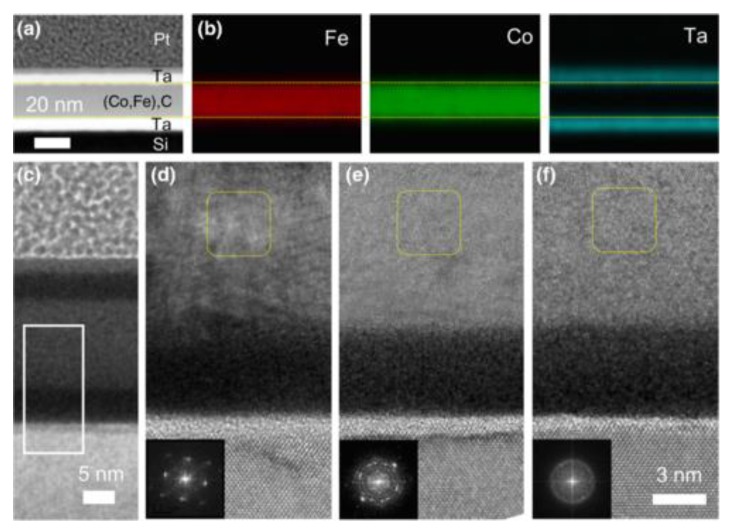
(**a**) A bright-field STEM image of Ta/(Co,Fe)C/Ta/SiO2/Si (001) substrate. (**b**) The corresponding energy-dispersive x-ray spectroscopy mappings for Fe, Co, and Ta elements, respectively. (**c**) Cross sections of lowresolution TEM with white solid rectangle displaying the corresponding location of HRTEM. (**d**–**f**) HRTEM images of Co-Fe-C alloy films with carbon content 0%, 4.4%, and 13.2%, respectively. The FFT of selected zones (yellow dotted squares) are inserted at the bottom left corner. Reproduced with permission from [112]. Copyright [2019], American Physical Society.

**Figure 15 sensors-20-01532-f015:**
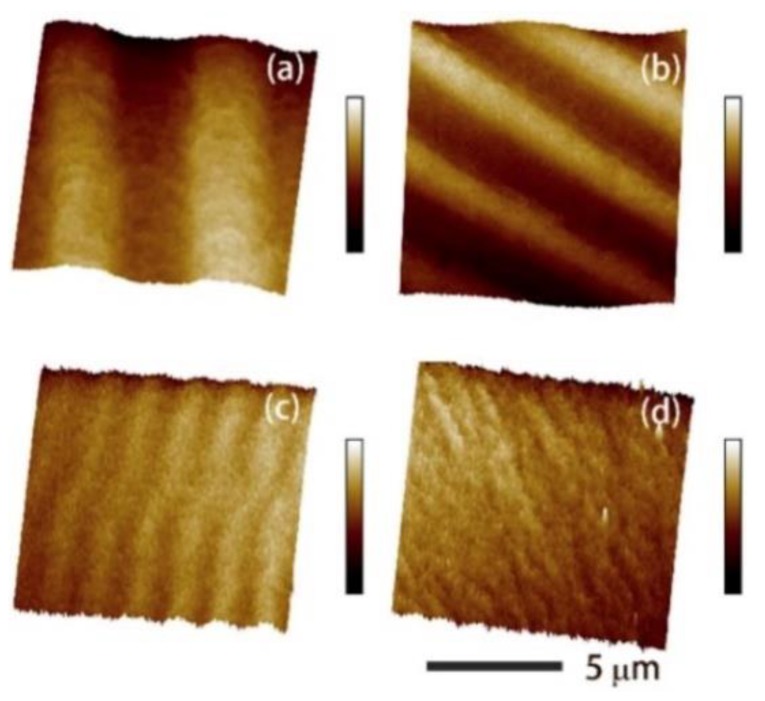
(**a–d**) are the surface magnetic domains of (480 nm Fe-Ga), (30 nm Fe-Ga/30 nm Fe-Ni)_8_, (16 nm Fe-Ga/16 nm Fe-Ni)_15_ and (8 nm Fe-Ga/8 nm Fe-Ni)_30_ films. Reprinted with permission from [114]. Copyright [2019], Elsevier.

**Figure 16 sensors-20-01532-f016:**
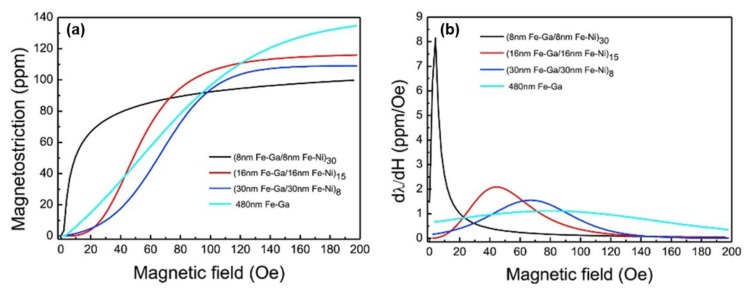
(**a,b**) are λ-H and dλ/dH-H curves of the samples, λ_s_ of the other samples are 137 ppm, 108 ppm,115 ppm and 97 ppm, d_33,m_ of the (8 nm Fe-Ga/8 nm Fe-Ni)_30_ reaches 8.1ppm/Oe. Reprinted with permission from [114]. Copyright [2019], Elsevier.

**Figure 17 sensors-20-01532-f017:**
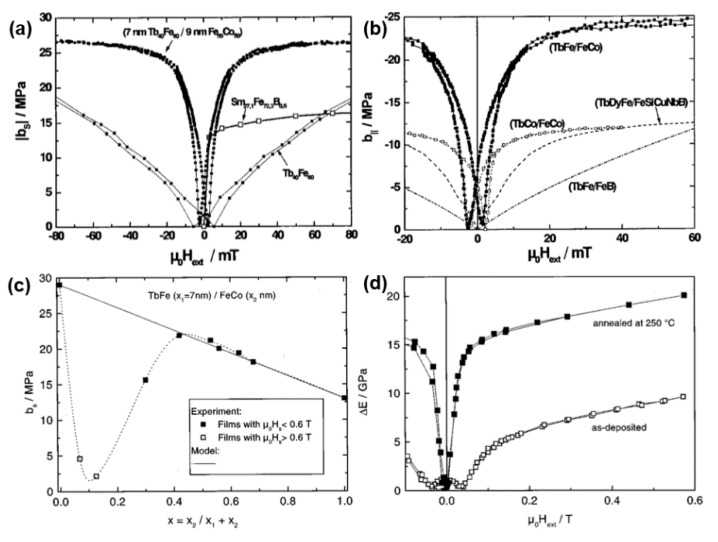
(**a**) Comparison of the magnetoelastic coupling coefficient of 250 °C annealed (7 nmTb_40_Fe_60_/9 nmFe_50_Co_50_) × 70 multilayer with an optimized Tb_40_Fe_60_ and a SmFeB film. (**b**) Comparison of the magnetoelastic coupling coefficients of (Tb_40_Fe_60_/Fe_50_Co_50_), (Tb_18_Co_82_/Fe_75_Co_25_), (TbDyFe/FeSiCuNbB) and (Tb_33_Fe_67_ /Fe_80_B_20_) multilayer films. (**c**) Saturation magnetoelastic coupling coefficient of TbFe/FeCo multilayers as a function of the soft magnetic layer thickness. (**d**) ΔE effect of as-deposited and at 250 °C annealed (7 nm TbFe/8 nm FeCo) multilayers. There figures are reproduced with the permission from [116,117]. Copyright [1999] and [2000], American Physical Society.

**Figure 18 sensors-20-01532-f018:**
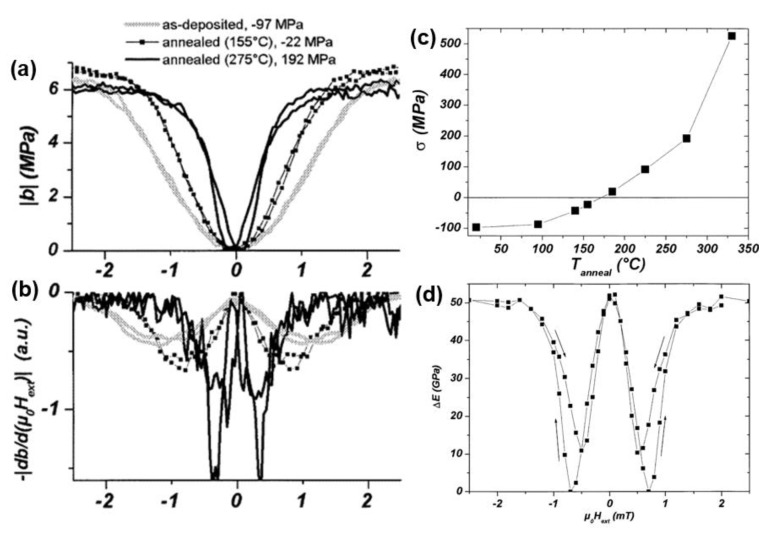
(**a**) Magnetoelastic coupling coefficient (**b**) negative absolute value of the first derivative of the magnetostrictive hysteresis (**c**) stress as a function of annealing temperature (**d**) ΔE effect of an optimized annealed (Fe_90_Co_10_)_78_Si_12_B_10_ thin film. Reproduced with permission from [118]. Copyright [2002], IEEE.

**Figure 19 sensors-20-01532-f019:**
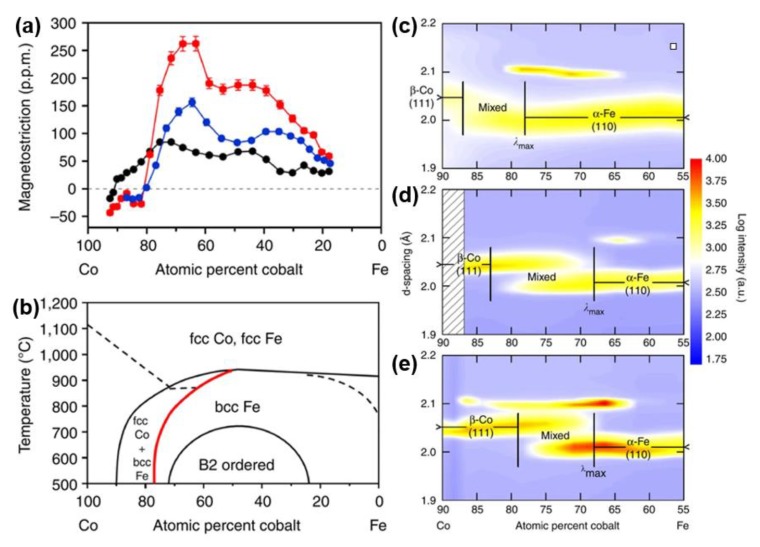
(**a**) Magnetostriction variation versus atomic percent cobalt for three differently prepared Co_1−x_Fe_x_ composition spreads, as-deposited (black dots), slow-cooled (blue dots), quenched (red dots). (**b**) Co–Fe-phase diagram. The red curve highlights the approximate phase boundary between (fcc Co + bcc Fe) and bcc Fe. (**c**–**e**) Intensity plots of the synchrotron microdiffraction of Co_1−x_Fe_x_ thin films. (**c**) as-deposited, (**d**) annealed and slow-cooled, (**e**) annealed and water-quenched composition spread samples. The black line marked λ_max_ in each spread indicates the approximate composition of the (fcc + bcc)/bcc phase boundary. Reprinted with permission from [119]. Copyright [2011], Springer Nature.

**Figure 20 sensors-20-01532-f020:**
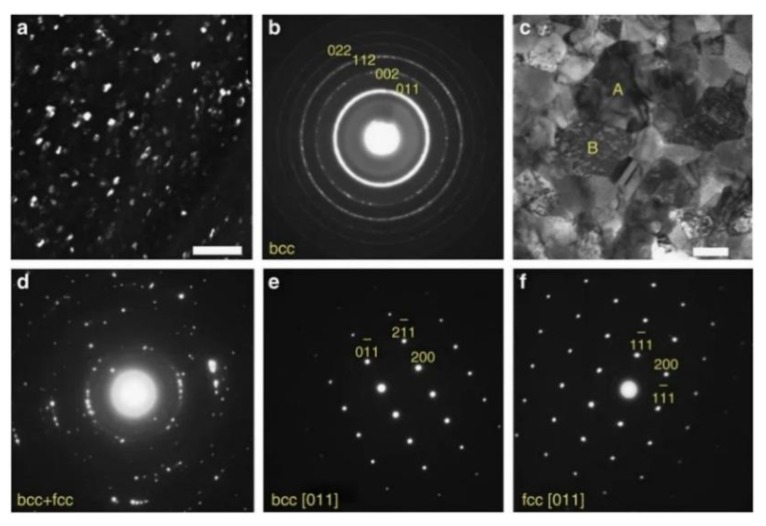
TEM images of Co0.73Fe0.27 of (**a**) (110) dark field image of as-deposited sample, the scale bar is 50 nm. (**b**) corresponding SAED pattern of the as-deposited, (**c**) bright field image of annealed sample, the scale bar is 200 nm. (**d**) SAED pattern of the same sample as (**c**) using ~1.5-µm-diameter aperture showing the mixture structure of bcc and fcc phases, (**e**) [11] bcc diffraction pattern from grain marked ‘A’ in (**c**), and (**f**) [11] fcc diffraction pattern from grain marked ‘B’ in (**c**). Reprinted with permission from [119]. Copyright [2011], Springer Nature.

**Figure 21 sensors-20-01532-f021:**
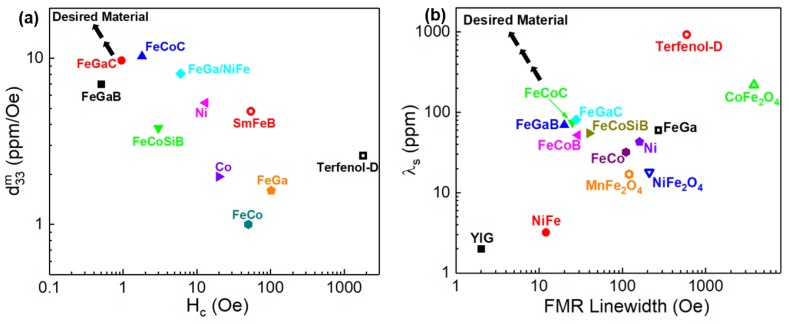
Magnetostrictive materials in summary of (**a**) maximum piezomagnetic coefficient d_33,m_ vs coercive field H_c_ (**b**) saturation magnetostriction λ_s_ vs FMR linewidth.

**Table 1 sensors-20-01532-t001:** Categories of various multiferroic devices. Adapted from [4].

ME Coupling	Physical Mechanisms	ME Devices	References
Direct ME coupling	H control of P	Magnetic/current sensors, energy harvesters, gyrators, transformers	[15,16,17,18,19,20,21,22,23,24,25]
Converse ME coupling	E control of M switching	MERAM	[26,27,28]
	E control of μ	Voltage tunable inductors, filters, phase shifters	[29,30,31,32,33,34,35,36]
Direct and converse ME coupling	Interaction between electric and magnetic phases	VLF mechanical antennas Nanomechanical antennas	[37,38,39,40,41,42] [43,44,45,46]
No ME coupling	High μ and ε	Compact antennas, etc.	[47,48]

Note: E/H, electric/magnetic field; P/M, polarization/magnetization; μ/ε, permeability/permittivity; MERAM, ME random access memory; VLF, very low frequency.

**Table 2 sensors-20-01532-t002:** Parameters of typical piezoelectric and magnetostrictive materials.

Parameters	Piezoelectric Phase					Magnetostrictive Phase				
	PZT-5H	PZT-8	PMN-0.33PT	LiNbO_3_	AlN film	Metaglas	Terfenol-D	FeGaC	FeGaB	FeCoSiB
**d_31,p_(pC N^−1^)**	−265	−37	−1330	−1	−2					
**d_33,p_(pC N^−1^)**	585	225	2820	21	~3.5–4					
$\varepsilon_{r}$	3400	1000	8200	30	~10					
**Q_m_**	65	1000	100	100000	500					
$\lambda_{s}$ **(ppm)**						~30	2000	81.2	70	158 * annealed
**d_33,m_(nm A^−1^)**						50.3	25	121.3	~88	
$\mu_{r}$						45000	10		~400	
**T_C_ (°C)**	193	300	135	1200	>2000	395	650			
**References**	[53]	[54]	[55]	[56]	[57,58,59]	[60]	[8]	[61]	[62]	[63]

Note: d_31,p_/d_33,p_, transverse/longitudinal piezoelectric constant; d_33,m_, longitudinal piezomagentic constant; λ_s_, saturated magnetostriction coefficient; ε_r_, relative dielectric constant; µ_r_, relative permeability; T_c_, curie temperature; Q_m_, mechanical quality factor; * The value is estimated using Young’s modulus and Poisson ratio of the film being 100 GPa and 0.3, respectively.

**Table 3 sensors-20-01532-t003:** Classifications of magnetoelastic effects.

Magnetoelastic Effects	
Direct Effects	Inverse Effects
**Joule magnetostriction** Change of dimensions in the direction of applied magnetic field	**Villari effect** Change of magnetization due to applied stress
**Volume magnetostriction** Change of volume due to spontaneous magnetization	**Nagaoka-Honda effect** Change of magnetization due to volume change
ΔE**effect** Dependence of Young’s modulus on the state of magnetization	Magnetically induced changes in the elasticity
**Wiedemann effect** Torque induced by helical anisotropy	**Matteuci effect** Helical anisotropy and electric and magnetic fields induced by a torque

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
