# Peer review of "A Review of Thin-Film Magnetoelastic Materials for Magnetoelectric Applications"

_sensors, 2020, doi:10.3390/s20051532_

Round 1

Reviewer 1 Report

A minor point, please define LOD.

There is no need to show how the number of papers has grown in recent years. Keep that for your funding agency.

The authors briefly describe some ME devices and conclude that the largest effects are observed in heterostructures consisting of piezoelectric films interspersed with magnetostrictive layers. They then go on to discuss in detail thin films of magnetostrictive alloys. There is no need to discuss so thoroughly thin film deposition processes. The discussion must be more balanced with a discussion not only of the magnetostrictive films but also of the piezoelectric films that are used with them. It would be interesting to consider one system and relate in detail the two components used and how they interact.

Reviewer 2 Report

The subject of the review is interesting and this work may attract special attention of the wide audience. Work is reasonable size for Review article and it is sufficiently well illustrated. At the same time, it does not cover all aspects of the research direction related to the thin-film magnetoelastic materials, existing applications and methods of such materials characterization. For example, flexible substrates are widely requested for different kind of magnetic sensors, including materials for high frequency applications (Appl. Phys. Lett. 2006, 88, doi:10.1063/1.2191950; Sensors 2014, 14, 7602-7624; doi:10.3390/s140507602; IEEE TRANSACTIONS ON MAGNETICS, VOL. 53, NO. 4, APRIL 2017, Digital Object Identifier 10.1109/TMAG.2016.2629513; Appl. Phys. Lett., vol. 64, no. 19, pp. 2593–2595,1994; etc.).

FMR line width in FeNi as well as coercivity were discussed for decades, including case for flexible substrates, I believe that some of related concepts must be added to the discussion.

It would be also an advantage to include 1-2 paragraphs with prospects of the development of existing research direction and view on possible novel applications.
